# Serving Graph Compression for Graph Neural Networks

**Si Si[1], Felix Yu[1], Ankit Singh Rawat[1], Cho-Jui Hsieh[2], Sanjiv Kumar[1]**
[1]Google Research
[2]University of California, Los Angeles
{sisidaisy, felixyu, ankitsrawat, sanjivk}@google.com
{chohsieh}@ucla.cs.edu

## Abstract

Serving a GNN model online is challenging — in many applications when testing nodes are connected to training nodes, one has to propagate information from training nodes to testing nodes to achieve the best performance, and storing the whole training set (including training graph and node features) during inference stage is prohibitive for large-scale problems. In this paper, we study graph compression to reduce the storage requirement for GNN in serving. Given a GNN model to be served, we propose to construct a compressed graph with a smaller number of nodes. In serving time, one just needs to replace the original training set graph by this compressed graph, without the need of changing the actual GNN model and the forward pass. We carefully analyze the error in the forward pass and derive simple ways to construct the compressed graph to minimize the approximation error. Experimental results on semi-supervised node classification demonstrate that the proposed method can significantly reduce the serving space requirement for GNN inference.

## 1 Introduction

Graph Neural Networks (GNNs) (Kipf & Welling, 2016) have been widely used for graph-based applications, such as node property predictions (Kipf & Welling, 2016), link predictions (Zhang & Chen, 2018), and recommendation (Wu et al., 2020). Given a graph that encodes relationships between pairs of entities, the graph convolution operation in GNN iteratively refines entity representations by aggregating features from neighbors, which enables information to propagate through the graph and boosts the performance on uncertain nodes.

It has been well recognized that GNN training on large-scale input graphs is challenging, and many scalable training methods have been proposed (Hamilton et al., 2017; Chiang et al., 2019; Chen et al., 2018a; Zeng et al., 2019). However, the problem of how to efficiently **serve** a GNN model in online applications remain unsolved. In fact, for applications when testing nodes are connected with the training nodes, such as semi-supervised node classification, serving a GNN is very challenging and has hindered the deployment of GNN in real world. To conduct predictions on a batch of incoming testing nodes, GNN has to propagate information not only within the testing nodes but also from training nodes to testing nodes, which implies that the serving system needs to store graph and node features in memory. Unfortunately, it is almost impossible to store the training graph and node features in many real systems such as embedded and low-resource devices. For example, for the Reddit dataset with more than 150k training nodes, it needs 370.7MB storage for the training node features, 86.0MB for the graph, and only 7.6MB for the GNN model itself. In Table 1, we break down the serving space requirements for four public datasets (with statistics in the experiment section) with GNN model size, training graph space, and training node features size. From this table, we can see that the space bottleneck is on the training graph and node features. We define the problem of reducing the size of training graph and node features as **serving graph compression** for GNNs.

Unfortunately, naively discarding all or a large portion of training nodes will significantly hurt the inference performance, since GNN has extracted very powerful node representations of training nodes, and propagating those information to testing nodes is crucial for prediction. Figure 1 illustrates

Table 1: Model size and serving space of several GNN models.

| Datasets | Model size | Training graph size | Training feature size | Total serving size |
|---|---|---|---|---|
| Arxiv | 1.4MB | 5.9MB | 46.5MB | 53.8MB |
| Reddit | 7.6MB | 86.0MB | 370.7MB | 464.3MB |
| Product | 4.8MB | 87.2MB | 78.6MB | 170.6MB |
| Amazon2M | 3.0MB | 485.4MB | 684.0MB | 1.17GB |

this problem, where we show that when discarding some part of training data, the performance of GNN will significantly reduce even on a standard node classification task.

Although the problem of serving graph compression has not been formally studied before, at the first glance the problem seems to be solvable by adopting existing approaches. The first approach one can easily come up with is to treat training node features as a weight matrix and apply existing model compression techniques such as sparse or low-rank approximation (Han et al., 2015; Frankle & Carbin, 2018; Sainath et al., 2013; Chen et al., 2018b). However, existing model compression methods are not able to exploit graph information, and we will show in the experiments that they tend to perform poorly for serving graph compression. Another straightforward idea is to treat the problem as a core-set selection problem, where many algorithms have been developed to select a subset of important samples from massive data (Mirzasoleiman et al., 2020; Wang et al., 2018; Zhao et al., 2020). However, core-set selection methods are trying to obtain a small subset such that the model trained on the subset still achieves reasonable performance, and this goal is different from serving

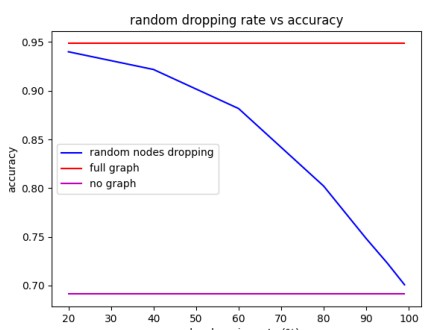

Figure 1: Random nodes dropping rate vs. accuracy on Reddit dataset. The red line is the accuracy of inference using the entire graph and features; purple line is the accuracy of inference without graph; blue line is accuracy for randomly dropping nodes in the graph with different dropping rate.

compression. For example, Jin et al. (2021) showed it's possible to extract a small synthesized graph for training, but they still require the whole training set in the inference phase to achieve good performance.

In this paper, we propose a simple and effective method for GNN serving graph compression via a virtual nodes graph (VNG). Given a GNN model to be served online, our method aims to construct a small set of **virtual nodes**, with artificially designed node features and adjacency matrix, to replace the original training graph. Without changing the model and the forward pass, users can just replace the original training set by the small representative set in the serving time to reduce the space requirement with small loss in testing accuracy. To construct the set of virtual nodes, we decompose the error of forward propagation into (1) the propagation error from training node features to testing nodes and (2) the propagation error from training to training nodes. Interestingly, the error in (1) can be bounded by a weighted kmeans objective, while the error in (2) can be minimized by solving a low-rank least-square problem to preserve consistency within a certain space constraint. These together lead to simple yet effective GNN serving graph compression that is easy to use in practice.

Our work makes the following contributions:

- To the best of our knowledge, this is the first work on the serving graph compression problem for GNN models, addressing the bottleneck when applying GNN in real world applications.
- By analyzing the error in forward propagation, we design a simple yet effective algorithm to construct a small virtual node set to replace the original huge training set for model serving.
- We show on multiple datasets that the proposed method significantly outperforms alternative ways of compressing the serving graph.

## 2 RELATED WORK

**Graph Neural Networks**    In this paper, we focus on GNN problems when there is a graph connecting entities of training and testing instances, including many important applications such as node classification, edge classification and recommendation systems. Note that there's another application of GNNs where each instance is a graph and we aim to predict properties of a new instance (graph), such as for molecular property prediction (Gilmer et al., 2017; Wu et al., 2018). Since in those applications training and testing graphs (instances) are disjoint and there is no need to store training graphs for serving, they are out of the scope of this paper.

Many improvements have been made on GNNs in recent years. Based on the graph convolution operation, many different designs of graph neural networks have been proposed to achieve better performance (Veličković et al., 2017; Shi et al., 2020; Li et al., 2021; Chen et al., 2020; Xie et al., 2020; Chien et al., 2021). Also, although training GNNs with large networks is difficult, many scalable training algorithms have been developed (Hamilton et al., 2017; Chen et al., 2018a; 2017; Chiang et al., 2019; Zeng et al., 2019). In contrast to previous works, we study the inference phase of GNN which has been overlooked in the literature. In the serving phase of GNN, it requires storing both GNN model weights and training data (including node features and training graph) to achieve the best performance, since connections from training to testing nodes are crucial for feature propagation. Since the size of training data is often more than 40x larger than GNN model weights, we focus on shrinking the size of training data in this paper. Existing works on compressing GNNs focus on reducing or quantizing the model weights (Tailor et al., 2020; Yang et al., 2020; Kim et al., 2021), which is orthogonal to our work. Another recent work (Chen et al., 2021) tried to prune both model weights and graph by removing some edges. However, they do not reduce the size of training features so the serving space requirement will not be significantly reduced.

**Model compression**  Since the training feature matrix dominates the serving space, a straight-forward method is to apply existing model compression techniques, such as pruning or low-rank approximation (Han et al., 2015; Frankle & Carbin, 2018; Sainath et al., 2013; Chen et al., 2018b), to the training feature matrix. However, we find those methods are not very effective in practice (see our experiments). This is because that standard compression methods to not consider feature matrix and graph jointly, and they do not really focus on reducing the forward pass error in GNN inference. Another commonly used compression method is quantization (Gholami et al., 2021). However, quantization is orthogonal to our method since we can also apply quantization after getting the virtual representative nodes with our method.

**Coreset selection and Dataset distillation**  Since the bottleneck of GNN serving space is training data, it may seen natural to apply coreset selection or dataset distillation methods here. Various coreset selection algorithms, including $k$-median (Har-Peled & Mazumdar, 2004), mixture models (Lucic et al., 2017), low-rank approximation (Cohen et al., 2017), and gradient matching methods (Mirzasoleiman et al., 2020), are designed to select a subset of essential samples to expedite training. On the other hand, dataset distillation methods (Wang et al., 2018; Zhao et al., 2020; Cui et al., 2022) create a small set of synthetic data that enables models trained on the synthetic dataset to achieve comparable performance to those trained on the original set. However, these methods do not utilize the graph information. Furthermore, these methods are designed to select a coreset for efficient training, while our goal is to select a coreset to compress the serving space requirement. We will compare with some standard coreset selection methods in the experiments to demonstrate that they are not effective for our task. Further, although  several recent works try to select coresets for graph data (Jin et al., 2021; Zhang et al., 2021), they still focus on selecting a coreset for training while utilizing the full training set in the inference phase. It's nontrivial to apply them for inference speed up since the connections between testing nodes and core-set nodes are not well defined.

**Graph Coarsening**  Graph coarsening is widely used in the scalable graph clustering, with the aim to reduce the size of the graph or form some supernodes in the graph to represent the original graph (Liu et al., 2018; Chevalier & Safro, 2009). Most of them are trying to preserve some unsupervised properties of the graph, such as adjacency matrix or normalized Laplacian (Spielman & Teng, 2011; Bravo Hermsdorff & Gunderson, 2019). Graph coarsening method do not consider either model or the node features, and thus are not as effective for our task.

## 3  PROPOSED METHOD

### 3.1  PROBLEM DEFINITION

For simplicity, we will consider the canonical graph convolutional network (GCN) model (Kipf & Welling, 2016), but our method can also handle other GNN architectures. In the training phase, we are given the training graph $G = (\mathcal{V}, \mathcal{E})$ with $n = |\mathcal{V}|$ vertices and $|\mathcal{E}|$ edges. The corresponding training adjacency matrix, denoted as $A_{tr,tr} \in \mathbb{R}^{n \times n}$, is a sparse matrix with $(A_{tr,tr})_{ij} \neq 0$ if and only if there is an edge between node $i$ and node $j$, and the value of $(A_{tr,tr})_{ij}$ denotes the weight of the edge. Note that standard GNN models usually conduct row normalization to the binary adjacency matrix and we directly assume $A_{tr,tr}$ is the normalized matrix here. Each node in the training graph is associated with a $d$-dimensional feature $\boldsymbol{x}_i \in \mathbb{R}^d$, and we use $X_{tr} \in \mathbb{R}^{n \times d}$ to denote the

feature matrix. We use $X_{i.}$ to denote the $i$-th row of a matrix $X$, so following this notation we have $(X_{tr})_{i.} = \boldsymbol{x}_i$. Starting with $X^{(0)} = X_{tr}$, an $L$-layer GCN propagates node features layer-by-layer:

$$Z^{(\ell+1)} = A_{tr,tr}X^{(\ell)}W^{(\ell)}, \quad X^{(\ell+1)} = \sigma(Z^{(\ell+1)}), \quad \ell = 0, \ldots, L-1. \tag{1}$$

where $X^{(\ell)}$ is the feature matrix at the $\ell$-th layer, $W^{(\ell)}$ is the weight matrix of a fully connected layer, and $\sigma$ is nonlinear activation (usually element-wise ReLU). $A_{tr,tr}X^{(\ell)}$ is the graph convolution operation, which obtains a new representation of each node by a weighted average of its neighbors.

After $L$ GCN layers, each node will obtain its final layer representation $X^{(L)}$, which will be associated with a loss function depending on the final task. A canonical application for GCN is semi-supervised node classification. In this application, each labeled node is associated with an observed label $y_i$, and we can use the standard cross-entropy loss to train the model. After training, we obtain the model weights $\mathcal{W} = \{W^{(0)}, \ldots, W^{(L-1)}\}$.

In this paper, we focus on the **serving phase** of the GNN models. Depending on different scenarios, testing nodes can come one-by-one, batch-by-batch, or all together. For simplicity, we assume conducting GCN inference on a single node $v$ with feature $\boldsymbol{x} \in \mathbb{R}^d$ and its connection to training nodes is represented by $\boldsymbol{a} \in \mathbb{R}^n$. In this case, GNN will conduct the following forward propagation to obtain node features

$$\begin{bmatrix} Z_{tr}^{(\ell+1)} \\ \boldsymbol{z}^{(\ell+1)} \end{bmatrix} = \begin{bmatrix} A_{tr,tr} & 0 \\ \boldsymbol{a} & \bar{a} \end{bmatrix} \begin{bmatrix} X_{tr}^{(\ell)} \\ \boldsymbol{x}^{(\ell)} \end{bmatrix} W^{(\ell)}, \quad \begin{bmatrix} X_{tr}^{(\ell+1)} \\ \boldsymbol{x}^{(\ell+1)} \end{bmatrix} = \sigma(\begin{bmatrix} Z_{tr}^{(\ell+1)} \\ \boldsymbol{z}^{(\ell+1)} \end{bmatrix}), \quad \boldsymbol{x}^{(0)} = \boldsymbol{x}, \tag{2}$$

where $\bar{a}$ is the constant depending how the GNN deals with the diagonal entries[1] Note that this can be easily generalized to cases with larger batch size $m$, where $\boldsymbol{x}$ will be replaced by an $m$-by-$d$ feature matrix; $\boldsymbol{a}$ will become an $m$-by-$n$ matrix representing the connections between testing and training nodes, and $\bar{a}$ will become the adjacency matrix between testing nodes.

We have demonstrated in Figure 1 that including training graph and node features in the serving phase is necessary. Intuitively, intermediate features for training nodes are very informative and propagating those information to testing nodes will significantly boost the performance. Also note that in (2) we assume there is no information flow from testing to training (0 in the top-right corner of joint adjacency matrix). This is because training nodes already have good features obtained in the training phase, so there is no need to propagate testing features to training nodes. This is adopted in several real GNN implementations (Chiang et al., 2019) and we found that setting the top-right corner of $A$ as 0 or $\boldsymbol{a}^T$ have similar performance in practice.

As evidenced from (2), the serving phase of GNN requires storing model weights $\mathcal{W}$, training node features $X_{tr}$ and training graph $A_{tr,tr}$. Storing $X_{tr}$ and $A_{tr,tr}$ requires $O(nd)$ and $O(|\mathcal{E}|)$ space respectively, and both of these are much larger than the size of $\mathcal{W}$ which consists of $L$ $O(d^2)$-sized matrices, with the corresponding storage requirement being independent of $n$. Taking a standard Reddit dataset as example, there are $n = 153,932$ training samples and $d = 602$ features, $X_{tr}$'s size is 370.7MB, $A_{tr,tr}$'s size is 86.0MB, and $\mathcal{W}$ is only 7.6MB size.

Accordingly, for GNNs, we define the task of reducing the space requirement for serving by primarily compressing $X_{tr}$ and $A_{tr,tr}$ as the **serving graph compression** problem.

## 3.2 THE VIRTUAL NODE GRAPH (VNG) METHOD

We propose the virtual node graph (VNG) method in this section. The main idea is to replace $n$ training nodes by a small number of $c$ **virtual (VR) nodes**, with $c \ll n$. Let $X_{vr} \in \mathbb{R}^{c \times d}$ be the features and $A_{vr,vr} \in \mathbb{R}^{c \times c}$ be the adjacency matrix of the VR nodes, we aim to obtain those matrices by minimizing the forward propagation error. With the training nodes replaced by VR nodes, the graph convolution operation (first equation of (2)) will be replaced by

$$\begin{bmatrix} Z_{vr}^{(\ell+1)} \\ \boldsymbol{z}^{(\ell+1)} \end{bmatrix} = \begin{bmatrix} A_{vr,vr} & 0 \\ \boldsymbol{a}_{vr} & \bar{a} \end{bmatrix} \begin{bmatrix} X_{vr}^{(\ell)} \\ \boldsymbol{x}^{(\ell)} \end{bmatrix} W^{(\ell)}. \tag{3}$$

Note that $\boldsymbol{a}_{vr}$ in (3) represents the connection from node $v$ to the VR nodes, which cannot be a fixed vector since it will depend on the the original $\boldsymbol{a}$ (the connections between node $v$ and training nodes).

---

[1]Can be 0 or 1 depending on the implementation, while some others methods add small constants to diagonal.

We thus assume each original training node is represented by one of the VR nodes, and the mapping is defined as $\pi : \{1, \ldots, n\} \to \{1, \ldots c\}$. We thus have

$$\boldsymbol{a}_{vr} = \boldsymbol{a}\,\Pi, \tag{4}$$

where $\Pi \in \{0,1\}^{n \times c}$ is the assignment matrix of the bipartite graph between training nodes and VR nodes. Specifically, $\Pi$ is a 0/1 matrix with $\Pi_{i,j} = 1$ if and only if $\pi(i) = j$.

With these setups, VR nodes will involve in two computations when computing the features for a given node $v$: 1) The propagation from VR nodes to $v$ 2) The propagation between VR nodes which will impact testing nodes in the next GNN layer. In the following, we will consider these two parts separately: in Sec 3.2.1 we consider 1) to obtain a solution for $\Pi$ and $X_{vr}$, and in 3.2.2 we consider the 2) to obtain a solution for $A_{vr,vr}$. We show with this two-step decomposition, all the components can be solved by simple closed form solutions which leads to an efficient and easy-to-implement algorithm. Although the solution can be potentially improved by solving the whole problem jointly, we leave it to the future work.

### 3.2.1 Propagation from virtual representative nodes to testing nodes

Propagation from VR nodes to node $v$ is the bottom row of the forward propagation in (3) and (2), where we want the resulting $\boldsymbol{z}^{(\ell+1)}$ to have minimal error. Therefore we want to have $\boldsymbol{a}X_{tr}^{(\ell)}W^{(\ell)} \approx \boldsymbol{a}_{vr}X_{vr}^{(\ell)}W^{(\ell)} = \boldsymbol{a}\Pi X_{vr}^{(\ell)}W^{(\ell)}$ which can be achieved if

$$\boldsymbol{a}X_{tr}^{(\ell)} \approx \boldsymbol{a}\Pi X_{vr}^{(\ell)}. \tag{5}$$

Since the testing nodes are unseen in the compression phase, we hope the compressed model to at least preserve the latent embedding when given every training node. If $v$ is the $i$-th training node, then $\boldsymbol{a} = (A_{tr,tr})_{i,\cdot}$. Therefore, based on the spirit of empirical risk minimization, we aim to construct the $\Pi$ matrix to minimize the following averaged square error:

$$\sum_{i \in [n]} \|(A_{tr,tr})_{i\cdot}X_{tr}^{(\ell)} - (A_{tr,tr})_{i\cdot}\Pi X_{vr}^{(\ell)}\|_2^2 = \sum_{i \in [n]} \| \sum_{j \in [n]} (A_{tr,tr})_{ij}(X_{tr}^{(\ell)})_{j\cdot} - (A_{tr,tr})_{ij}(X_{vr}^{(\ell)})_{\pi(j)\cdot}\|_2^2$$

$$\leq \sum_{i \in [n]} \sum_{j \in [n]} (A_{tr,tr})_{ij}\|(X_{tr}^{(\ell)})_{j\cdot} - (X_{vr}^{(\ell)})_{\pi(j)\cdot}\|_2^2$$

$$= \sum_{j \in [n]} (\sum_{i \in [n]} (A_{tr,tr})_{ij})\|(X_{tr}^{(\ell)})_{j\cdot} - (X_{vr}^{(\ell)})_{\pi(j)\cdot}\|_2^2. \tag{6}$$

Let $\gamma_j = \sum_{i \in [n]}(A_{tr,tr})_{ij}$ being the column sum of $A_{tr,tr}$, finding the VR nodes' features $X_{vr}^{(\ell)}$ and mapping $\pi(\cdot)$ is then equivalent to the weighted kmeans objective, with weight $\gamma_j$ for training node $j$.

As we aim to minimize the error for all layers, we can bound the sum of (6) over all layers. If we define the concatenation of features from all layers as

$$\bar{X}_{tr} = [X_{tr}^{(0)}, \ldots, X_{tr}^{(L-1)}], \tag{7}$$

since the weights are independent to layers, we can lower bound the sum of loss over layers per (6):

$$\sum_{\ell \in [L]} \sum_{i \in [n]} \|(A_{tr,tr})_{i\cdot}X_{tr}^{(\ell)} - (A_{tr,tr})_{i\cdot}\Pi X_{vr}^{(\ell)}\|_2^2 \leq \sum_{j \in [n]} v_j\|(\bar{X}_{tr})_{j\cdot} - (\bar{X}_{vr})_{\pi(j)\cdot}\|_2^2. \tag{8}$$

As the right hand side is equivalent to the weighted kmeans objective on $\bar{X}_{tr}$, we can apply the well-known EM-style weighted kmeans algorithm to (approximately) optimize the upper bound, where it iteratively updates the cluster centers $\bar{X}_{vr}$ and the cluster assignment $\pi$. After running weighted kmeans, we will obtain the assignment $\pi^*$, and the cluster centers are weighted average of training features within each cluster, so ideally we would like VR nodes' features at layer $\ell$ to be

$$X_{vr}^{(\ell)} = EX_{tr}^{(\ell)}, \text{ where } E_{ij} = \begin{cases} \frac{v_j}{\sum_{q:\pi^*(q)=i} v_q} & \text{if } \pi(j) = i, \\ 0 & \text{otherwise.} \end{cases} \tag{9}$$

Intuitively, $E$ can be viewed as edges from training nodes to VR nodes, where each training node is only linked to one VR node. Since we only have freedom to assign the input features for virtual representative nodes, we set $X_{vr}^{(0)} = EX_{tr}^{(0)}$, and since beyond first layer the features are computed by propagating previous layers feature with $A_{vr,vr}$, we will discuss how to assign $A_{vr,vr}$ to best preserve this property for other layers. Note that we assume all layers share the same $\Pi$, which means each virtual node in different layers correspond to the same set of training nodes. Having different $\Pi$ for each layer requires different $A_{vr,vr}$ for each layer, significantly increasing the space complexity.

### 3.2.2 PROPAGATION BETWEEN VIRTUAL REPRESENTATIVE NODES

Note that the $E$ matrix obtained by weighted kmeans captures the transformation from training nodes to virtual nodes, and we have enforced (9) in the input layer. We will then study how to set the propagation among VR nodes or *equivalently* define the matrix $A_{vr,vr}$ such that it preserves the property of (9) for other layers.

Towards this, assume that $X_{vr}^{(\ell)} = EX_{tr}^{(\ell)}$ holds at layer $\ell$. We then want to make sure that conducting a graph convolution on both VR nodes and training nodes will preserve this relationship, i.e.,

$$A_{vr,vr}X_{vr}^{(\ell)} = A_{vr,vr}EX_{tr}^{(\ell)} \approx EA_{tr,tr}X_{tr}^{(\ell)}, \tag{10}$$

where the left hand side is the propagation of VR nodes and the right hand side is the propagation of original training nodes, with an additional projection operator to map the features back to VR nodes.

Note that we want the above approximation to hold for all layers $\ell \in [0, L-2]$. Thus, summing the approximation error for each layer leads to the following optimization problem:

$$A_{vr,vr} = \arg\min_H \|HE\bar{X}_{tr} - EA_{tr,tr}\bar{X}_{tr}\|_F^2, \tag{11}$$

where $\bar{X}$ is defined in (7). Note that (11) can be easily solved by a standard least square solver. However, this will lead to a dense $c \times c$ matrix, which leads to storage bottleneck when $c$ is large. Therefore, we propose to obtain a low-rank solution of (11) by requiring $H$ to have a rank at most $c < k$. Such a low-rank matrix can then be stored with only $O(ck)$ memory overhead. With the rank constraint our optimization problem to solve for $A_{vr,vr}$ takes the form:

$$A_{vr,vr} = \arg\min_H \|HP - Q\|_F^2 \quad \text{s.t.} \quad \text{rank}(H) \leq k, \tag{12}$$

where $P = E\bar{X}_{tr}$ and $Q = EA_{tr,tr}\bar{X}_{tr}$. We now present a closed-form optimal solution for (12).

**Theorem 1** *Assume $P = U_p\Sigma_pV_p^T$ and $Q = U_Q\Sigma_QV_Q^T$ are the thin-SVD of $P$ and $Q$, respectively. Then the solution of (12) is $(QV_p)_k\Sigma_p^{-1}U_p^T$, where $(\cdot)_k$ denotes the rank-$k$ truncated SVD of a matrix.*

The proof is deferred to the appendix. With this closed form solution, it is straightforward to solve (12). Note that the closed form solution requires to compute the thin-SVD of $P$ and $Q$. These are $c$-by-$d$ matrices where $c$ is the number of virtual nodes and $d$ is the hidden dimension size (sum over all layers). As $d$ is usually small (e.g., less than $3,000$ in all our cases) and the complexity of SVD is $O(\min(c, d)^2 \max(c, d))$, thin-SVD will be efficient (see Appendix C for details). In practice, this can be done within 1 minutes on standard CPU, even without GPU. Note that it is necessary to store rank-$k$ matrix $A_{vr,vr}$ in a factorized form to realize the aforementioned $O(ck)$ storage cost during inference, and our closed form solution in the above theorem naturally provide such a factorization.

### 3.3 OVERALL ALGORITHM

Here we summarize the proposed Virtual Node Graph (VNG) algorithm from Section 3.2. Given the original GNN model and training data, we will first compute the collection of training node features before graph convolution $\bar{X}_{tr} = [X_{tr}^{(0)}, \ldots X_{tr}^{(L-1)}]$. For general GNN, these are features before one-hop propagation or mathematically, left-multiplying with the adjacency matrix. We then compute the column sum of the $A_{tr,tr}$ to get $\{\gamma_1, \ldots, \gamma_n\}$; subsequently we run weighted kmeans on $\bar{X}_{tr}$ using these weights (cf. (8)). Given the clustering assignment $\pi$ obtained from the weighted kmeans, we form an $E$ matrix based on (9) and set features for VR nodes as $X_{vr}^{(0)} = EX_{tr}^{(0)}$. We then solve (12) with $P = E\bar{X}_{tr}$ and $Q = EA_{tr,tr}\bar{X}_{tr}$ to obtain $A_{vr,vr}$ (stored in the factorized form). At the inference time, the forward propagation is conducted by (2), where we only store virtual nodes instead of training nodes. The algorithm reduces the space complexity from the original $O(n + |\mathcal{E}| + nd)$ to $O(ck + cd)$, where $c \ll n$ is the number of virtual representative nodes and $k$ is a low-rank factor in (12). Please see Algorithm 1 for the description of the algorithm.

## 4 EXPERIMENTAL RESULTS

To evaluate the performance of the proposed compression algorithm, we conduct experiments on four real datasets:

---

**Algorithm 1** The Virtual Node Graph (VNG) algorithm

---

**Input**: Training adjacency matrix $A_{tr,tr}$ and features $X_{tr}$, GNN weights $\mathcal{W}$, number of virtual nodes $c$, rank of virtual adjacency matrix $k$.

1: Compute $\bar{X}_{tr} = [X_{tr}^{(0)}, \ldots, X_{tr}^{(L-1)}]$ by GNN forward pass with training data $A_{tr,tr}$ and $X_{tr}$.
2: Compute weights $\gamma_j = \sum_i (A_{tr,tr})_{ij}$ for all $j$.
3: $\pi \leftarrow$ weighted_kmeans$(\bar{X}_{tr}, c, \{\gamma_i\}_{i=1}^n)$.
4: Get $E$ matrix based on (4).
5: Compute $X_{vr} = EX_{tr}$ ($X_{tr}$ is the raw feature for training nodes).
6: Solve $A_{vr,vr}$ based on (12).

**Output**: $A_{vr,vr}$ and $X_{vr}$

---

Table 2: The statistics of Arxiv, Reddit, Product, and Amazon2M datasets.

| Datasets | #Training Nodes | #Validate Nodes | #Labels | #Features | Serving size |
|---|---|---|---|---|---|
| Arxiv | 90,941 | 29,799 | 40 | 128 | 52.4MB |
| Reddit | 153,932 | 23,699 | 41 | 602 | 456.7MB |
| Product | 196,615 | 39,323 | 47 | 100 | 165.8MB |
| Amazon2M | 1,709,997 | 739,032 | 47 | 100 | 1.17GB |

- **Arxiv**: A multi-class node classification dataset used for predicting the category of a paper. We use the same dataset and partition as in (Hu et al., 2020).
- **Reddit**: A multi-class node classification dataset used for predicting the communities of online posts. We use the same dataset and partition as in (Chiang et al., 2019).
- **Product**: A multi-class classification dataset that is similar to Amazon2M , with a key difference that it is based on a different preprocessing and split by Hu et al. (2020).
- **Amazon2M**: A multi-class classification dataset, where the graph has products co-purchase history, data features have each product's information, and the label is the category of the product. We use the same dataset and partition as in (Chiang et al., 2019).

All above datasets are publicly available and are commonly used for benchmarking the performance of GNNs on node classification tasks. The statistics of these datasets are summarized in Table 2. As this is the first paper considering the serving graph compression problem for GNN, we mainly compare our method against standard model compression methods (for compressing training node features) and core-set selection methods (selecting a subset of nodes for serving). Specifically, we include the following methods in the comparison:

- **Sparse**: Sparsifying the training nodes' features by sorting the values and removing small values in the features. For this method, the connection graph does not change. The serving space is controlled by the pruning ratio.
- **SVD**: Performing SVD over the training nodes' features to save the serving space. The serving space is controlled by the rank used in performing low-rank approximation via SVD. Also, note that the original graph remains unchanged for this method.
- **Random**: Randomly selecting some training nodes and their corresponding subgraph.
- **Degree**: Selecting the training nodes by sorting the degree of training nodes and picking the nodes with the highest degrees.
- **Kmeans**: Performing kmeans over the training nodes features, and selecting the training nodes closest to the kmeans centroids.
- **VNG** (our proposed method): Constructing the virtual nodes graph which is much smaller than the original graph based on the GNN forward pass reconstruct error (cf. Algorithm 1).

We use the vanilla GCN architecture used in Chiang et al. (2019) to test the above methods. Note that the proposed compression algorithm can also be used for other GNN architectures, such as GraphSage Hamilton et al. (2017) (see Appendix E). We use ClusterGCN's tensorflow implementation [2] for GCN model training. As for architecture, on all datasets, we consider a 4-layer GCN model with hidden dimensions 512, 256, 512, and 400 for Product, Arxiv, Reddit, and Amazon2M, respectively, and the mean aggregator from Hamilton et al. (2017). To test each model, we first train a GCN model and freeze the model, then we run each method to generate a compressed graph for efficient

---

[2]https://github.com/google-research/google-research/tree/master/cluster_gcn

inference on the new testing nodes. The process of graph compression and subsequent inference with the compressed graph does not involve retraining the GCN model.

Here we consider two settings of inference. The first setting is batch inference, where we are given a batch of testing nodes at a time, with their node features and interconnections. The connection from the testing nodes to the training nodes are also given. This is the typical setting for GNN model inference. The second setting is single node inference, where we assume the testing nodes appear one-by-one in an on-line fashion. In this setting, we only know the testing node features and the connections from this testing node to the training nodes.

## 4.1 BATCH INFERENCE SETTING

As explained previously, in the batch inference setting, we assume the testing data become available in batches. Since the edges connecting training nodes to testing nodes and the connections among testing nodes are given, each testing node's features will be updated using the information from the training nodes and other testing nodes. We show the compression results in Table 3, where the graph size (including nodes' features and adjacency graph size) is shown in brackets following the accuracy. We construct the compressed graph with only 1% and 5% of the total number of training nodes for our method. Note that Kmeans, Degree, and Random generate the graphs that have the same number of nodes as VNG.

From Table 3, we observe that our method is able to achieve a very high compression rate for the serving space – 37.5x on Arxiv, 41.1x on Reddit, and 69.1x on Product. Other methods will suffer from a significant performance loss when using such a compression rate. Note that Sparse and SVD both operate on nodes' features, while maintaining the original adjacency graph. Thus, the compressed graph is still as large as the original graph. Random is not aware of the information in the graph and features. Kmeans only uses node features and is not aware of adjacency graph information. Degree considers adjacency graph and does not take features into consideration. Therefore, all the methods except ours consider either adjacency graph or node features, but not both. Furthermore, these methods are model agnostic as the GNN model itself does not play a role in the compression. In contrast, our method considers all three factors: adjacency graph, node features, and GNN model; consequently, performing better than other baselines.

Besides reducing the inference serving size, VNG can also speed up the inference. The detailed discussions and results about inference speed of VNG can be found in Table 8 in the appendix. Also in Appendix A, we conduct an ablation study to show different variations of our method such as replacing weighted kmeans with vanilla kmeans or choosing different ranks during low-rank step.

## 4.2 SINGLE NODE INFERENCE SETTING

In the single node inference setting, each testing node appears one-by-one. Unlike the batch setting, no connections among testing nodes are given. Therefore, how to effectively utilize the information from training data becomes even more crucial. We compare our method with all the baselines in this setting and present the results in Table 4. Since our method is able to better utilize the information of training graph and features, the accuracy gap between our method and baselines becomes larger.

## 4.3 TRADEOFF BETWEEN COMPRESSION RATE AND ACCURACY

In this subsection, we vary the compression methods in a wider range to demonstrate the tradeoff between compression ratio and accuracy. We consider the single node inference setting and plot the accuracy under different compression rates for each method in Figure 2. On each dataset, we increase the compression rate of VNG until its accuracy is within 1% of the full model, and show the performance of other methods under the same compression rate. It can be easily observed that our method significantly outperforms other methods also in this high-accuracy regime. The actual numbers for the accuracy and model size can be found in Table 9 in the appendix.

## 5 CONCLUSION

Although GNNs achieve promising performance handling graph data, they require storing training graph and features to perform inference, leading to a high serving space requirement. In this paper, we propose a novel method for serving graph compression, where we replace the training features and graph by a small set of virtual nodes. The features and adjacency matrix of virtual nodes are designed to minimize the forward propagation error of GNN and can be simply adapted to any GNN

Table 3: Accuracy% (Size) on four datasets for batch setting where test data appears in batches. The brackets next to accuracy show the compressed graph size including both adjacency matrix and node features. 'Full' corresponds to using the original adjacency graph and nodes features. For 'Random', 'Degree', 'Kmeans', 'VNG', we show the results under different compression rates in the 'Node Compression rate' column–1% or 5% means the compressed graph contains 1% or 5% of the total number of nodes of the original graph. *For Amazon2M the number of nodes in the compressed graph is 0.1% or 0.5% of the original graph.

| Model | Node Compression rate | Arxiv | Reddit | Product | Amazon2M* |
|---|---|---|---|---|---|
| Full | - | 69.94 (52.4MB) | 94.85 (456.7MB) | 87.51 (165.8MB) | 89.15 (1.17GB) |
| SVD | - | 58.69 (8.1MB) | 88.35 (104.5MB) | 40.82 (91.1MB) | 68.08 (512.7MB) |
| Sparse | - | 56.90 (6.0MB) | 68.27 (93.4MB) | 64.75 (88.7MB) | 66.45 (486.1MB) |
| Random | 1% | 58.32 (2.3MB) | 71.23 (11.2MB) | 70.26 (2.5MB) | 77.80 (2.1MB) |
| | 5% | 62.85 (12.0MB) | 77.91 (57.5MB) | 74.57 (13.7MB) | 77.98 (10.4MB) |
| Degree | 1% | 63.63 (3.2MB) | 76.00 (15.5MB) | 73.81 (3.8MB) | 78.99 (2.8MB) |
| | 5% | 66.42 (14.8MB) | 86.62 (78.1MB) | 79.36 (24.5MB) | 81.71 (38.4MB) |
| Kmeans | 1% | 58.09 (2.4MB) | 71.93 (11.4MB) | 69.73 (2.5MB) | 77.75 (2.1MB) |
| | 5% | 61.93 (12.0MB) | 80.31 (60.3MB) | 72.83 (14.3MB) | 78.33 (21.0MB) |
| VNG (ours) | 1% | 67.07 (1.4MB) | 90.51 (11.1MB) | 83.11 (2.4MB) | 83.12 (2.1MB) |
| | 5% | 68.64 (7.0MB) | 91.00 (55.6MB) | 85.60 (11.8MB) | 84.23 (10.2MB) |

Table 4: Accuracy% (Size) on four datasets for single setting where testing data appears one-by-one setting. In the brackets next to accuracy, we show the compressed graph size including both adjacency matrix and node features. 'Full' corresponds to using the original adjacency graph and nodes features. For 'Random', 'Degree', 'Kmeans', 'VNG', we show the results under different compression rates in the 'Node Compression rate' column–1% or 5% means the compressed graph contains 1% or 5% of the total number of nodes of the original graph. *For Amazon2M, #nodes in the compressed graph is 0.1% or 0.5% of original graph.

| Model | Node Compression rate | Arxiv | Reddit | Product | Amazon2M* |
|---|---|---|---|---|---|
| Full | - | 68.35 (52.4MB) | 92.75 (456.7MB) | 86.95 (165.8MB) | 87.79 (1.17GB) |
| SVD | - | 55.37 (8.1MB) | 81.46 (104.5MB) | 39.64 (91.1MB) | 54.72 (512.7MB) |
| Sparse | - | 51.54 (6.0MB) | 54.55 (93.4MB) | 60.81 (88.7MB) | 61.32 (486.1MB) |
| Random | 1% | 52.71(2.3MB) | 56.62 (11.2MB) | 66.15 (2.5MB) | 61.01 (2.1MB) |
| | 5% | 58.57 (12.0MB) | 69.22 (57.5MB) | 71.18 (13.7MB) | 61.34 (10.4MB) |
| Degree | 1% | 60.58 (3.2MB) | 62.58 (15.5MB) | 7013 (3.8MB) | 63.53 (2.8MB) |
| | 5% | 63.95 (14.8MB) | 77.18 (78.1MB) | 76.77 (24.5MB) | 67.64 (38.4MB) |
| Kmeans | 1% | 52.51 (2.4MB) | 57.42 (11.4MB) | 65.57 (2.5MB) | 60.99 (2.1MB)) |
| | 5% | 57.75 (12.0MB) | 67.53 (60.3MB) | 69.31 (14.3MB) | 61.94 (21.0MB) |
| VNG (ours) | 1% | 66.73 (1.4MB) | 90.51 (11.1MB) | 83.04 (2.4MB) | 82.53 (2.1MB) |
| | 5% | 67.84 (7.0MB) | 91.40 (55.6MB) | 85.08 (11.8MB) | 84.26 (10.2MB) |

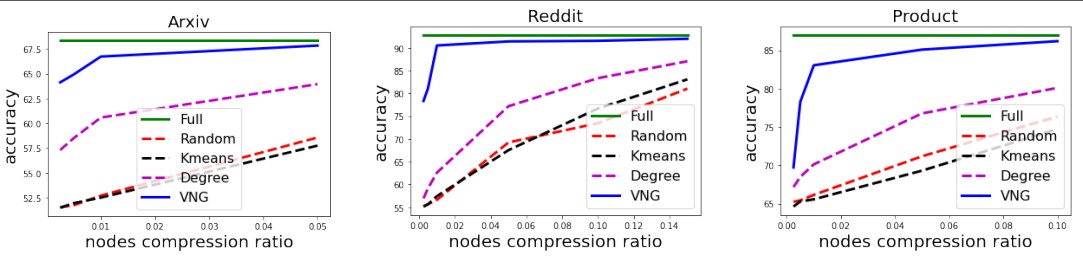

Figure 2: Comparisons of graph compression for GCN inference methods (single node inference setting). The x-axis shows the node compression ratio. The y-axis shows the accuracy of the node classification tasks. The graph sizes for Sparse and SVD are too large and they cannot realize the compression ratios considered in the figure.

structure. Our empirical results demonstrate that the proposed method can significantly reduce the GNN serving size while maintaining a reasonable predictive performance. As one of the limitations of our method, it may suffer from higher performance loss if a larger compression rate (e.g., 50x) is desired, but it is possible to further improve by combining it with pruning or quantization approaches.

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
