# OpenReview forum: "Serving Graph Compression for Graph Neural Networks"
_ICLR.cc/2023/Conference — ICLR 2023 notable top 25%_

### Official Review · Reviewer_LGG4 · 2022-10-23

**Confidence:** 3
**Correctness:** 3
**Technical Novelty And Significance:** 3
**Empirical Novelty And Significance:** 2
**Recommendation:** 6

**Clarity, Quality, Novelty And Reproducibility:**

Clarity
- The problem setup and motivation are clear. The paper also provides details about its methodology.

Quality
- The writing quality of this paper overall is pretty decent and is easy to follow.

Novelty
- The paper has good discussions in the introduction and related work about its novelty.  Although compression has been an extensively studied topic, compressing the feature vectors for GNN with virtual node sets appears to be novel.

Reproducibility
- The paper does not provide many details about its implementation and hyperparameter settings for most of its configurations, which creates difficulties reproducing its results.


**Strength And Weaknesses:**

Strengths:
- The paper makes some interesting observations that when serving GNN models, nodes in the training sets may still be required, which leads to increased memory consumption.
- The paper introduces the virtual node set, which decouples the memory consumption of serving GNN from the training node features.
- Promising accuracy improvements in comparison to alternative baseline methods.

Weaknesses:
- Despite claiming memory is a major bottleneck in GNN serving, most datasets evaluated in this paper are rather small scale, with <500MB of memory consumption except Amazon 2M (<1.2GB), which are much smaller than the DRAM capacity. Also, it does not seem like all training features need to be loaded into DRAM at once, e.g., by loading the testing nodes and their neighbors or by loading a subset of training nodes on the fly, the memory usage can also be reduced.
- Despite showing accuracy improvement in comparison to baselines created by the authors, the absolute accuracy drop in comparison to the uncompressed GNN is still large. For example, the accuracy drop is around 5 points for Amazon 2M. In practice, it is unclear whether such a huge accuracy drop can justify the compression ratio. Also, would other methods perform better than the proposed method at a higher accuracy target? One suggestion is to decrease the compression ratio to find a configuration that leads to similar or at least within <1% of accuracy loss compared with the uncompressed GNN and then reports the compression ratio the proposed method can achieve. Furthermore, it would be interesting to report under different compression ratios, how the proposed method compares with other alternatives.
- Lack of comparison with quantization methods. The authors compared many methods, such as pruning and SVD. However, from the reviewer's perspective, one straightforward and also effective compression method is just to quantize the feature vectors,  e.g., basic quantization schemes using an INT8 symmetric quantizer. Would that lead to better accuracy than the proposed method (e.g., at 25% of compression ratio)?


**Summary Of The Paper:**

This paper studies the problem of reducing the memory consumption of serving GNN models. In particular, it identifies that some GNN models require node features in the training set data to make accurate predictions, which can add significant memory pressure. To mitigate this issue, the paper introduces VNG, a method that replaces some training nodes with a small set of so-called virtual nodes. By doing so, the total amount of memory consumption for serving GNN is reduced. Evaluation of GCN and several datasets show that the proposed methods can lead to higher accuracy than alternative methods.


**Summary Of The Review:**

Despite there being some concerns about the evaluation of the proposed methods, I'm overall positive about the paper because of the identification of the challenges of serving GNN models and the promising results of achieving a high compression ratio while obtaining better accuracy than alternative methods.

---

> ### Author Response · Authors · 2022-11-19
> **Response to Reviewer LGG4 -- Part 1**
>
> Thank you for taking the time to review our submission. We are glad that the reviewer appreciates the novelty and importance of the problem identified in our paper; and found our solution and experimental results promising. Please see below for a point-by-point response to your main concerns.
>
> Q1:  Despite claiming memory is a major bottleneck in GNN serving, most datasets evaluated in this paper are rather small scale, with <500MB of memory consumption except Amazon 2M (<1.2GB), which are much smaller than the DRAM capacity. Also, it does not seem like all training features need to be loaded into DRAM at once, e.g., by loading the testing nodes and their neighbors or by loading a subset of training nodes on the fly, the memory usage can also be reduced.
>
> Ans: Thank you for this good question. We’d like to emphasize that GNN inference does not only require loading testing nodes and their neighbors. To compute the prediction on a testing node at layer L, it requires that node’s neighbor nodes’ embeddings at layer L-1, which again needs their neighbors’ embeddings at layer L-2 and recursive ones in the downstream layers. Therefore, the number of required nodes (or their embeddings) scales exponentially in L. Thus, computing a batch of testing nodes may require a large portion of training nodes. Further, the access pattern is very irregular so it will be very slow if we store training nodes in the lower level of the memory hierarchy.
>
> Note that our largest model, Amazon2M, requires 1.17GB when uncompressed. Although it can easily fit in DRAM, it’s not small compared to models in Vision and NLP. For instance, a standard ResNet-50 takes only around 100MB space; a BERT-base model takes only around 1.3GB space. And there have been hundreds of papers trying to compress those models. Model compression is also important in the regime of 1GB-2GB space due to the following reasons:
>
> 1: If we store the models on edge/mobile devices, it’s not desirable that a single model takes 1.17GB. Users wouldn’t want to download an app that uses 1.17GB just for storing a single model.
>
> 2: Further, in many edge devices the memory is limited, so >1 GB model size could be a huge burden.
>
> 3: It’s worth pointing out that, in web-level applications, there could be billions of nodes. So the serving space for GNN will be much larger than Amazon2M. The model will not be able to fit in GPU memory, and creating a large-memory CPU instance will also be costly.
>
> Q2: it would be interesting to report under different compression ratios, how the proposed method compares with other alternatives.
>
> Ans: Thanks for this suggestion. There is a trade-off between accuracy and compression rate. In Table 3 and Table 4 in the main text,  we compare different methods under the same set of node compression rates (e.g., 1%, 5%), and show our method can achieve much better accuracy under the same space constraint. As c (number of virtual nodes) increases, our method can also achieve accuracy that is close to the original model. To show this, we increase the compression rate of VNG until its accuracy is within 1\% of the full model, and compare it with other approaches. The results can be found in Figure 2 and Table 9 (see detailed descriptions in Section 4.3). For example, with node compression rates of 5% on Arxiv, 15% on Reddit, and 10% on Product datasets, we can achieve less than 1% accuray loss.  It can be easily observed that our method significantly outperforms other methods also in this high-accuracy regime.
>
> We have also plotted under different compression ratios, how the proposed method compares with other alternatives in Figure 2. The results indicate that our method dominates all other alternatives under all different compression ratios.

---

> ### Author Response · Authors · 2022-11-19
> **Response to Reviewer LGG4 -- Part 2**
>
> Q3: Lack of comparison with quantization methods.
>
> Ans: Thanks for this question. Firstly we want to point out that our method can be combined with quantization methods (quantized over the distilled node features and edge weights), which could be an orthogonal and interesting future direction. Secondly, we compared our method with INT8 symmetric quantizer and INT4 symmetric quantizer under the batch setting. And the results comparing full data v.s. our method (5% and 10% of nodes compression rate) v.s. INT8 symmetric quantizer v.s. INT4 symmetric quantizer are shown below. The number in the parentheses is the serving graph size (adjacency graph+nodes features). Our method is consistently better than INT4 symmetric quantizer in terms of accuracy and serving graph size. For comparison with INT8 symmetric quantizer, on arxiv dataset, our method has slightly lower accuracy (68.95% vs 69.92%), while with a better compression rate (13.9MB vs 17.5MB). On the reddit dataset, our method is better in accuracy (91.53% vs 91.26%) and compression rate (111.9MB vs 178.7MB). From the table, we can also observe a tradeoff between compression rate and accuracy.
>
> |        | Full            | Our method (5%) | Our method (10%) | INT8 symmetric quantizer | INT4 symmetric quantizer |
> |--------|-----------------|-----------------|------------------|--------------------------|--------------------------|
> | arxiv  | 69.94 (52.4MB)  | 68.64 (7.0MB)   | 68.95 (13.9MB)   | 69.92 (17.5MB)           | 62.58 (11.7MB)           |
> | reddit | 94.85(456.6MB) | 91.00 (55.6MB)  | 91.53 (111.9MB)  | 91.26 (178.7MB)          | 32.8 (132.3MB)           |

---

### Official Review · Reviewer_afao · 2022-10-24

**Confidence:** 3
**Correctness:** 3
**Technical Novelty And Significance:** 2
**Empirical Novelty And Significance:** 2
**Recommendation:** 3

**Clarity, Quality, Novelty And Reproducibility:**

Clarity and quality need to be improved;
Novelty moderate;
Reproducibility unclear as code link is not provided.


**Strength And Weaknesses:**

Strength
1. The setting for online GNN inference is described clearly.
2. The two algorithms for feature vector and graph structure compression make sense.
3. The empirical results are strong, having much higher accuracy than other methods at the same compression rate.

Weakness
1. The applications of online GNN inference are unclear. It would help if the authors could elaborate more on applications that use online GNN inference. My doubt is that the case of adding new nodes may be less common than adding/removing edges between existing nodes. For example, for recommendation and social networks, most of the nodes (e.g., users, products) may be already there but the edges experience constant changes due to user interaction. Also, in some business applications (e.g. PinSAGE), all embeddings in the graph are recomputed periodically.

2. Compression yields obvious accuracy loss in the experiments. This may be unacceptable for applications such as recommendation because even marginal accuracy loss is important to revenue. Using out-of-core solutions (e.g., storing some data on SSD) may be more favorable to avoid accuracy loss.

3. Experiments and presentations need to be improved. (1) Include some large graphs (e.g., MAG-240M and Papers100M) in the experiments, which helps to justify the need of compression; (2) Report the time used to compress the graphs (although analysis show that the complexity is low); (3) Adapt some graph coarsening methods (I think the task may be challenging but the results are good to have). For presentation, (1) Include the size of each dataset in Table 2; (2) Plot Figure 2 to make the axis and legends clearer; (3) On page 3, the authors mention that coreset selection has been used in training, what are the difficulties of adapting these methods for inference? Simply stating that these methods do not consider inference is not enough.


**Summary Of The Paper:**

This paper proposes to compress the graph and node features when serving the online inference of GNN. The scenario is that new nodes come with their connections to existing nodes, and output vectors of the new nodes need to be computed. The authors use a k-means based algorithm to compress the feature vectors, and an SVD-based algorithm to compress the graph structure. Empirical results show that the proposed method yields higher accuracy than other baselines.

**Summary Of The Review:**

The paper considers compressing graph data for GNN inference. However, the application scenarios need to be further justified and the performance loss is significant. Moreover, experiments can be improved.

---

> ### Author Response · Authors · 2022-11-19
> **Response to Reviewer afao -- Part 1**
>
> Thank you for your time to review our manuscript. We are happy to learn that the reviewer liked the problem setting and appreciated that our empirical results are stronger as compared to other baselines. Please see our detailed response to your comments/concerns below. We hope that our responses will prompt the reviewer to reassess our submission.
>
> Q1: The applications of online GNN inference are unclear. It would help if the authors could elaborate more on applications that use online GNN inference. My doubt is that the case of adding new nodes may be less common than adding/removing edges between existing nodes.
>
> Ans: There are many examples where we need GNNs to deal with newly added nodes. For instance, in product tagging or recommendation we need to deal with newly added products; in youtube tagging/recommendation we need to deal with newly added videos. As you mentioned in the review comment, pre-computing and storing the post-GNN final embeddings periodically could be one solution, but this also implies that they are unable to deal with newly added nodes in a timely manner. Many current systems are not able to conduct predictions in a dynamic way due to a lack of algorithms for light-weighted GNN serving, which is exactly the main problem we are trying to solve in this paper. Further, there are cases where the nodes are more dynamic (e.g., a query typed by a user, a new comment/question entered by a user, or a query image). One can imagine that any newly added Reddit post or reply is a node, and the prediction (e.g., recommendation to another Reddit article) has to be made dynamically. Also, Zhang et al., 2020 showed that by constructing a GNN with both testing images and database images, a better image representation can be obtained for the image retrieval task. In this application, testing images may not appear in the database so cannot be pre-computed.
>
> (Zhang et al., 2020) Understanding Image Retrieval Re-Ranking: A Graph Neural Network Perspective.
>
> Q2: Compression yields obvious accuracy loss in the experiments. This may be unacceptable for applications such as recommendation because even marginal accuracy loss is important to revenue. Using out-of-core solutions (e.g., storing some data on SSD) may be more favorable to avoid accuracy loss.
>
> Ans: Thank you for the suggestion. As c (number of virtual nodes) increases or nodes compression rate reduces, our method can also achieve accuracy that is close to the original model. To show this, we increase the compression rate of VNG until its accuracy is within 1\% of the full model, and compare it with other approaches. The results can be found in Figure 2 and Table 9 (please see detailed descriptions in Section 4.3). For example, with node compression rates of 5\% on Arxiv, 15\% on Reddit, and 10\% on Product datasets, we can achieve less than 1% accuray loss. It can also be easily observed that our method significantly outperforms other methods in this high-accuracy regime.
>
> Using an out-of-core approach to store embeddings can definitely be a solution but may suffer from slow inference speed. Even though an SSD is faster, it is still much slower (at least 20x slower) than direct memory access. Further, the access pattern of GNN inference is very irregular, since we need to access all L-hop nodes for a single test node. Therefore, it may be difficult to load embeddings in chunks from SSD to memory.
>
> Q3: Include some large graphs (e.g., MAG-240M and Papers100M) in the experiments.
>
> Ans: Thanks for all the suggestions. Firstly, we want to emphasize that our work is compressing a training graph given a pretrained model by the user. The efficient training of GCN models on huge scale data is out of scope of this work. To conduct experiments on those gigantic graphs, we have tried our best to find checkpoints for those larger models, however without any luck. Also, we considered training GCN models on these data from scratch, however, it turns out that it requires massive GPU memory or clusters for training as the graph itself cannot fit into regular GPU memory— paper100m itself is more than 85GB. Given the time constraints and the resource requirements for these big datasets, we are unfortunately not able to get a good GCN model on MAG-240M and Papers100M at the moment. We will consider them in our future work.

---

> ### Author Response · Authors · 2022-11-19
> **Response to Reviewer afao -- Part 2**
>
> Q4: Report the time used to compress the graphs (although analysis show that the complexity is low)
>
> Ans: Regarding the efficiency of forming the compressed graph (preprocessing time), our method is actually very efficient. We did not include too much detail in the original paper since preprocessing time, which can be conducted offline, is often less important for compression methods. We have added more details on the preprocessing time in Appendix C.
>
> Our method mainly consists of three steps: (1) Inference of the training data to obtain their embeddings; (2) performing weighted k-means (Section 3.1); and (3) performing low-rank approximation (Section 3.2).
>
> For step (1) we only need to perform the forward pass of the training data once to obtain the embeddings for each layer. For our largest Amazon2M dataset, it takes 11.1 seconds.
>
> For (2),  weighted k-means takes $O(ncdT)$ time for clustering, where $T$ is the number of steps for weighted-k-means, typically <20;  $c$ is the number of clusters (virtual nodes); $d$ is feature dimension and $n$ is number of training nodes. When $c$ is large, we can use hierarchical k-means so the complexity will be roughly $O(ndT*\log(c))$. In this step, we could also speed up the weighted k-means by sub-sampling of training data. For Amazon2M dataset with 0.5% compression rate, it takes 3234.8 seconds.
>
> For (3), as mentioned at the end of Section 3.2 and Appendix C, the computational complexity is $O(min(c,d)^2 max(c,d))$. Since d is usually small (e.g., less than 3000 in all our cases), this is very efficient. Detailed discussions can be found in Appendix C. For our largest Amazon2M dataset with 0.5\% compression rate, this step takes 44.4 seconds.
>
> In summary, on Amazon2M dataset the preprocessing time of our method takes around 3290 seconds while training a GNN on this dataset requires more than 1 day. Therefore applying our method will not introduce a large overhead.
>
> Q5: Adapt some graph coarsening methods
>
> Ans: We’d like to point out that graph coarsening cannot be directly used for GCN training, as the graph coarsening just compresses the graph, and node features are not presented in the graph coarsening process.  We follow the same procedure as in (Yu et al., 2017) for graph coarsening using Metis as graph clustering method. As for super nodes’ features,  we take the average of each cluster nodes’ features and use these features for GCN inference. We tested this method on Arxiv dataset under the same node compression rates.  The results are shown as below:
>
> | Node compression rate | VNG    | graph coarsening  |
> |-----------------------|--------|-------------------|
> | 1%                    | 67.07% | 63.39%            |
> | 5%                    | 68.64% | 64.38%            |
>
> We can see that our method achieves higher accuracy than graph coarsening method under the same node compression rate.
>
> (Yu et al., 2020) Graph Coarsening with Preserved Spectral Properties.
>
> Q6: For presentation, (1) Include the size of each dataset in Table 2; (2) Plot Figure 2 to make the axis and legends clearer
>
> Ans: Thanks for the suggestions. We have modified the table and figure in the new version.
>
> Q7: the authors mention that coreset selection has been used in training, what are the difficulties of adapting these methods for inference?
>
> Ans: (Jin et al., 2021; Zhang et al., 2021) proposed coreset selections on graph data but these cannot be used to speed up inference since the connections between testing nodes and core-set nodes are not defined. For instance, each node in (Jin et al., 2021) is a synthetic node that is not corresponding to any real node. We have added this to the paper.

---

> ### Author Response · Authors · 2022-11-28
> **Looking forward to your further comments**
>
> Dear Reviewer afao,
>
> Thank you very much again for your valuable and insightful comments and suggestions for this paper, which indeed helps us to improve this work. We hope our replies have addressed your concerns about our work. If you are satisfied with our reply, we would deeply appreciate it if you could raise your score. Although the deadline for discussion between reviewers and authors has passed, we are still looking forward to your further feedback and/or questions and keep improving our work.
>
> Many thanks,
>
> Authors

---

### Official Review · Reviewer_f4pk · 2022-10-25

**Confidence:** 4
**Correctness:** 4
**Technical Novelty And Significance:** 4
**Empirical Novelty And Significance:** 4
**Recommendation:** 8

**Clarity, Quality, Novelty And Reproducibility:**

I enjoyed the paper reviewering.
The manuscript is highly easy-to-read. The problem is stated clearly, proposes solutions step by step.

The huge trainng graph is problematic to deploy the trained GNN for not-so-powefull machines in the wild.
The manuscript identifies this problem and derives technically sound solutions.
The numbers in the experimental results strongly support the claims.

Perferably, comparison with quantizqation methods will make the paper even stronger. They may be orthogonal to the proposed method, but it is not easy in general to deploy two approaches at the same time.

The only concern is that the GNN reseraches broaden so much that I cannot follow all of its aspecets. Thus there is always a possibility that some prior works raised the similar questions in the past, though the reviewer is not aware of.

Some questions

* How can we deal with grpahs with multi-type edges? for example, bond types in atomic molecular graphs, like/list/buy  relations of user-item purchase records.
* The representations of the virtual nodes are confinedin the same space with the training node features. I guess this is due to the formulation of  Eq.(4). Is there a possibility that we can reduce the number of virtual nodes further, by allowing the virtual node features to have a larger dimensionality (>d) to augment its "Informative"-ness?


**Strength And Weaknesses:**

(+) the proposed problem is interesting, practically useful, and new to the community (At the reviewer's knowledge)

(+) the manuscript is well structured, easy to read, no logical jumps.

(+) the proposed solutions are simple and reasonable.

(+) experimental results.

(-) Tecnnical novely of the proposed solutions are not high: but this is a minor problem since the main virtue of the manuscript is the proposal of a new problem.

(-) preferably, compare with quantization methods.



**Summary Of The Paper:**

This paper proposes a new problem of serving (test-time/deploy-phase inference) of a trained GNNs efficiently.
Many GNN papers work on a semi-supervised node classification task. In that semi-supervised node classification scenarios, the trained GNN must retrain the huge memory load ot the entire training graphs to perform message passing of test nodes/edges.
The goal of this paper to reduce this test-time computatonal load in the test time infernece by compressing the training data nodes into a smaller set of virtual nodes. This is a practically valuable problem to tackle, but first introduced by this paper.

Each training nodes must be mapped one of these virtual nodes. The mapping is learnf by weighted K-means of the tranining nodes.
Propagation paths within the virtual nodes are also solved by a simple Frobenius norm minimization.

The experimental results show that the proposed method outperfroms naive methods for reducing test-time computatonl load in terms of the accuracy deteriorations from the "full" test prediction.


**Summary Of The Review:**

This paper proposes a new and pracitically valuable question of reducing the computational burden of GNNs.
The idea is to compress the huge training graph into a smaller virtual node graph, by solving two simple optimization problems.
The rationale is clear, proposed optimization problems seems reasonable.
Strong experimental results support the goodness of the proposd problem and the solution.

---

> ### Author Response · Authors · 2022-11-19
> **Response to Reviewer f4pk--Part 1**
>
> Thank you for reviewing our submission. We are glad that the reviewer found our proposed solution interesting and practically useful, and acknowledged that our experimental results strongly support the utility of the proposed solution. Below, we provide a detailed response to all of your comments/questions.
>
> Q1: Technical novely of the proposed solutions are not high: but this is a minor problem since the main virtue of the manuscript is the proposal of a new problem.
>
> Ans: Thank you for recognizing our main novelty on proposing a new GNN compression problem. However, we do want to emphasize that the proposed method, although looks simple, is nontrivial. First, we show that the propagation error can be bounded by weighted k-means, which is nontrivial. Further, we show that there’s a closed-form solution for the low-rank problem, which is different from standard SVD (Theorem 1). Therefore, although the algorithm is simple to implement, it is not a trivial extension of any existing method.
>
> Q2: preferably, compare with quantization methods.
>
> Ans: Thanks for this question. Firstly we want to point out that our method can be combined with quantization methods (quantized over the distilled node features and edge weights), which could be an orthogonal and interesting future direction. Secondly, we compared our method with INT8 symmetric quantizer and INT4 symmetric quantizer under the batch setting.  And the results comparing full data v.s. our method (5% and 10% of nodes compression rate) v.s. INT8 symmetric quantizer v.s. INT4 symmetric quantizer are shown below. The number in the parentheses is the serving graph size (adjacency graph+nodes features). Our method is consistently better than INT4 symmetric quantizer in terms of accuracy and serving graph size. For comparison with INT8 symmetric quantizer,  on arxiv dataset, our method has slightly lower accuracy (68.95% vs 69.92%), while with a better compression rate (13.9MB vs 17.5MB). On the reddit dataset, our method is better in accuracy (91.53% vs 91.26%) and compression rate (111.9MB vs 178.7MB). From the table, we can also observe a tradeoff between compression rate and accuracy.
>
> |        | Full            | Our method (5%) | Our method (10%) | INT8 symmetric quantizer | INT4 symmetric quantizer |
> |--------|-----------------|-----------------|------------------|--------------------------|--------------------------|
> | arxiv  | 69.94 (52.4MB)  | 68.64 (7.0MB)   | 68.95 (13.9MB)   | 69.92 (17.5MB)           | 62.58 (11.7MB)           |
> | reddit | 94.85(456.6MB)  | 91.00 (55.6MB)  | 91.53 (111.9MB)  | 91.26 (178.7MB)          | 32.8 (132.3MB)           |

---

> ### Author Response · Authors · 2022-11-19
> **Response to Reviewer f4pk--Part 2**
>
> Q3: How can we deal with graphs with multi-type edges? for example, bond types in atomic molecular graphs, like/list/buy relations of user-item purchase records.
>
> Ans: Our algorithm can be extended to deal with multi-type edges. Consider the R-GCN framework (Schlichtkrull et al., 2017) where there are R adjacency matrices $A^{(1)}, …, A^{(R)}$ denoting different edge types, they consider the forward propagation pass as below
>
> $Z^{(\ell)} = \sum_{r=1}^R A^{r}X^{(\ell)}W^{(\ell, r)}, \ \ X^{(\ell+1)}=\sigma (Z^{(\ell+1)})$
>
> where the main idea is to generate features for the r-th type by $X^{(\ell)}W^{(\ell, r)}$ and then propagate each of them through the corresponding adjacency matrix. Features at all r views are then aggregated into a single feature and passed to the next layer.
>
> Our method can be adapted to this setting, though some improvements may be possible to better deal with multi-relational edges. Our algorithm consists of two parts: (1) Using weighted k-means to find the virtual nodes’ features, and (2) Using the proposed low-rank approach to form the adjacency matrix between virtual nodes.
>
> For (1), since the features will propagate through r graphs, we can consider merging the loss of those r graphs in eq(5), so the loss in eq(6) becomes the sum of loss in all graphs, which can still be reduced to weighted k-means objective but with the weights ($\gamma_j$) being defined as the sum of the degree in all adjacency matrices ($\gamma\_j = \sum\_{r=1}^R\sum\_{i\in [n]}(A\_{tr,tr})\_{ij}^{(r)}$). There’s an approximation here where we ignore the different weight matrices for each r ($W^{r}$) here. Considering different weight matrices in this clustering objective can potentially lead to better results, but we leave this as a future direction.
>
> For (2), since we’ll need to define R adjacency matrices between virtual nodes, eq(10) will become $A\_{vr,vr}^{(r)}EX^{(\ell)}\_{tr}W^{(\ell, r)} \approx E A\_{tr,tr} X^{(\ell)}\_{tr}W^{(\ell, r)}$ for each $r=1, …, R$. Since those $A\_{vr,vr}^{(r)}$ are independent, we can just solve each of them by Eq(12) and Theorem 1 independently to get $R$ low-rank adjacency matrices between virtual nodes.
>
> Due to the time constraint, we are not able to conduct experiments on R-GCN, and it is not the focus of this paper. We believe that designing methods to carefully handle multiple edge types and conduct comprehensive experiments on different GNN models can be an interesting future work.
>
> Schlichtkrull et al., “Modeling Relational Data with Graph Convolutional Networks.” 2017.
>
> Q4: The representations of the virtual nodes are confined in the same space with the training node features. I guess this is due to the formulation of Eq.(4). Is there a possibility that we can reduce the number of virtual nodes further, by allowing the virtual node features to have a larger dimensionality (>d) to augment its "Informative"-ness?
>
> Ans: This is an interesting direction and may further improve our method. Currently, we do not have a good idea on how to extend the method to allow virtual nodes to have larger d, since this will make the feature dimension inconsistent with the weight matrices in the GNN model’s checkpoint which is trained beforehand. It could be an interesting future work to study how to change the GNN model to allow higher dimensional virtual nodes’ features.

---

> > ### Comment · Reviewer_f4pk · 2022-11-20
> > **thank you!**
> >
> > authors,
> >
> > thank you for answering my feedbacks!
> > I feel the answers are satisfactory, and am happy to offer a couple of future work topics.
> >
> > Currently I will keep my positive score.
> > I would like to report my final score at the last moment, after (possible) opinion exchangings among other reviewers.
> >
> >
> > Thanks!

---

### Official Review · Reviewer_t66n · 2022-10-25

**Confidence:** 4
**Correctness:** 4
**Technical Novelty And Significance:** 4
**Empirical Novelty And Significance:** 3
**Recommendation:** 8

**Clarity, Quality, Novelty And Reproducibility:**

**Clarity:**
Section 3 was, as mentioned above, a pleasure to read. The writing and mathematics were clear, accessible, and flowed well. The first two sections did a sufficient job of identifying key issues (e.g., compressing graph structure/features vs. weights) as well as identifying related approaches drawn from the graph literature outside of GNNs. Section 4 was less clear and suffered from a lack of messaging and poor plots.

**Quality:**
The core mechanism presented is sound, well-motivated, and clever. While there are a few generality questions, the work as-presented seems solid and the results (while limited) clearly show the method succeeds at its intended purpose.

**Novelty:**
The proxy graph constructed here is, as far as I am aware, novel. In addition, I consider the whole approach of the paper (using a smaller, synthetic, derived graph instead of attempting to compress the graph and features themselves) to be a useful shift in perspective. I believe both provide something to the community.

**Reproducibility:**
The method seems to be described in sufficient detail to be replicated without issue. I especially appreciated section 3.3's summary and listing 1. A companion code repository would be appreciated in the un-anonimzed final version, as the algorithm does not seem onerous to implement for the public.

**Strength And Weaknesses:**

**Strengths:**
- The mechanism presented in the paper is mathematically clever, clean, and (after being explained) intuitive. Instead of attempting to brute-force the solution via direct graph/feature compression (which has extensive literature and could certainly have been applied, albeit with worse results I'd expect), they came up with a mechanism that synergizes with the behavior of GCNs. It's to their credit that the method can be accurately described as 'optimizing for GCN loss under space constraints' instead of 'optimizing for space and hoping for the best'.

- The paper is incredibly well-structured. The pieces flow well from each other, and reading Section 3 was a surprisingly pleasant experience. The mathematics are accessible and clear.

- The problem addressed is understudied, and the solution presented seems novel.

**Weaknesses:**
- The solution depends heavily on the formulation of the GNN model used. I suspect that the method can be generalized to other GNNs (perhaps with less mathematical simplicity), but the fact remains that it's tied to one incarnation as presented in this paper.

- The problem setup is a bit artificial. While I understand the reason for using existing, widely-used datasets and classification tasks, I'm not convinced (as one who does not serve GNNs for inference as my day job) that the examples chosen are representative of the datasets and learning problems facing GNN serving. The authors don't seem to provide much support to the contrary, so it begs the question whether the choices are oversimplifying or skewing the results.

- The evaluation was a bit underwhelming compared to the rest of the paper. The writing was poorer and confusing in places, key pieces of information were hard to find, the graph are not particularly readable, and the parameter sweeps were undermotivated and narrow. I suspect some of this could be fixed prior to publication, but the differences between sections 3 and 4 were palpable in the current version. Also, I noticed a distinct lack of computational performance information (the single comment at the end of section 3.2 was all I could find), which is a bit concerning in a paper which is trading compute for space.

**Summary Of The Paper:**

In order to support GNN inference via a served, pre-trained model, the authors present a method for constructing a small, auxiliary graph data structure which can act as a proxy for the features and structure of a larger training graph. This behaves similarly to compressing a training graph and allows a smaller chunk of data to be passed to inference clients.

**Summary Of The Review:**

Overall, the method is clever and interesting. The paper does a good job of laying it out clearly for the reader, and the results are enough to convince me it works. There are some rough edges (generality, problem definition, evaluation) that could use improvement, but they don't seem bad enough to invalidate any of the main takeaways.

---

> ### Author Response · Authors · 2022-11-19
> **Response to Reviewer t66n**
>
> Thank you for your positive comments on the novelty and the clarity of our paper. We are also happy to know that the reviewer thinks the proposed algorithm is clever and clean. Below, we provide a detailed response to all of your comments/questions.
>
> Q1: generalized to other GNNs?
>
> Ans: The compression algorithm is designed to optimize the error in the neighborhood propagation step in GNN (a simple $A*x$ computation where A is the adjacency matrix and x is node features) and does not rely on any other component. Therefore, it can be easily applied to most of the GNN variants. For instance, we have reported the results on another GNN architecture, GraphSage, in the appendix (Table 7). Our method also achieves a high compression rate in that architecture.
>
> Q2: The problem setup is a bit artificial
>
> Ans: The datasets we used are the commonly used benchmark for GNN node classification tasks, and cover almost all the majority of node classification problems in Open Graph Benchmark (OGB) (Arxiv and Product) and some previous large-scale benchmarks used prior to OGB (Amazon2M and Reddit). Other node classification benchmarks, such as Cora, Citeseer, PPI are too small so we didn’t include them in the paper. Our collection of the datasets covers problems associated with reddit, arxiv papers, and Amazon product recommendation. We agree that there might be some particular task that crucially requires GNN compression, and we believe our method can be applied as well since it is quite problem-independent.
>
> Q3: The evaluation was a bit underwhelming compared to the rest of the paper.
>
> Ans: We have replotted the figures and revised Section 4 to improve the readability. Regarding the hyper-parameters, we are using the Cluster-GCN codebase and all the hyper-parameters of the GCN models are identical to the original paper [Chiang et al., 2019]. The proposed compression method only has two hyper-parameters: $c$ (number of virtual nodes) and $k$ (rank of the adjacency matrix $A_{{vr},{vr}}$ between virtual nodes). $c$ corresponds to the compression ratio. We set $c$ as 1% and 5% of the total training nodes (and also other values when plotting the compression rate v.s. accuracy curves) to demonstrate the performance under different compression ratios. Regarding $k$, as shown in the ablation section in Appendix A, there is a trade-off between graph size and accuracy when choosing the low-rank $k$ for approximating $A_{{vr},{vr}}$. In practice, we choose rank $k$=min(# of compressed nodes, feature dimension).
>
> Q4: lack of computational performance information
>
> Thank you for the advice. Due to the space limit, we included the inference time analysis and actual running time for the proposed method in the appendix (Appendix G and Table 8). From those results, our method is faster than the baseline and does not sacrifice inference speed. In particular, for the largest dataset (Amazon2M), our method achieves a 4x inference time speedup.
>
> Regarding the efficiency of forming the compressed graph (preprocessing time), our method is actually very efficient. We did not include too much detail in the original paper since preprocessing time, which can be conducted offline, is often less important for compression methods. We have added more details on the preprocessing time in Appendix C.
>
> Our method mainly consists of three steps: (1) Inference of the training data to obtain their embeddings; (2) performing weighted k-means (Section 3.1); and (3) performing low-rank approximation (Section 3.2).
>
> For step (1) we only need to perform the forward pass of the training data once to obtain the embeddings for each layer. For our largest Amazon2M dataset, it takes 11.1 seconds.
>
> For (2),  weighted k-means takes $O(ncdT)$ time for clustering, where $T$ is the number of steps for weighted-k-means, typically <20;  $c$ is the number of clusters (virtual nodes); $d$ is feature dimension and $n$ is number of training nodes. When $c$ is large, we can use hierarchical k-means so the complexity will be roughly $O(ndT*\log(c))$. In this step, we could also speed up the weighted k-means by sub-sampling of training data. For Amazon2M dataset with 0.5% compression rate, it takes 3234.8 seconds.
>
> For (3), as mentioned at the end of Section 3.2 and Appendix C, the computational complexity is $O(min(c,d)^2 max(c,d))$. Since d is usually small (e.g., less than 3000 in all our cases), this is very efficient. Detailed discussions can be found in Appendix C. For our largest Amazon2M dataset with 0.5% compression rate, this step takes 44.4 seconds.
>
> In summary, on Amazon2M dataset the preprocessing time of our method takes around 3290 seconds while training a GNN on this dataset requires more than 1 day. Therefore applying our method will not introduce a large overhead.

---

### Public Comment · ~Linfeng_Cao1 · 2023-05-04
**Public Code**

Dear authors, thank you so much for presenting such a great work! I'm interested in this graph compression area as well and want to conduct some empirical analysis at beginning. I'm wondering is there any code publicly available to implement your method?  I'll be very appreciated if you could provide some code for an easier implementation. Thank you so much!

---

### Decision · Program_Chairs · 2023-01-20

**Decision:**

Accept: notable-top-25%

**Justification For Why Not Higher Score:**

The fundamental idea of compression by reducing the no of nodes is quite common.

**Justification For Why Not Lower Score:**

 The confident reviewers were very excited about the paper. The proposal is simple and seems quite effective.

**Metareview: Summary, Strengths And Weaknesses:**

The paper proposed a neat and clear strategy to compress the graph for serving GCNN model during inference. The idea is to create a small graph with significantly less no of nodes such that GCNN with node embedding trained on original graph can directly be used on the new graph with modified node embedding and the same inference algorithm.  This is a clear possibility and does solve a very real problem of reducing the churn during inference.

The experiments are rigorous with all the interesting baselines. The fact that a clean method outperforms significantly many principled ideas is amazing.

**Note From Pc:**

if the above contains the word "oral" or "spotlight" please see: "oral" presentation means -> notable-top-5% and "spotlight" means -> notable-top-25%. As stated in our emails, we are disassociating presentation type from AC recommendations

**Summary Of Ac-Reviewer Meeting:**

Reviewers were very excited about the results, clever idea, neat presentation and simplicity of the algorithm in practice. One of the reviewers has some concerns  about the settings, which the area chair found to be not a problem. Reducing inference time cost is a big deal.  Also the reviewer never participated in the discussions, so the reviewers scores were ignored.